# CONSTRAINED VARIATIONAL GENERATION FOR GENERALIZABLE GRAPH LEARNING

## ABSTRACT

Out-of-distribution (OOD) generalization aims at dealing with scenarios where the test data distribution can largely differ from training data distributions. Existing works for OOD generalization on graphs generally propose to extract invariant subgraphs that provide crucial classification information even under unseen test distributions. However, such a strategy is suboptimal due to two challenges: (1) *intra-graph correlations*, i.e., correlated structures that are partial invariant, and (2) *inter-graph distinctions*, i.e., significant distribution shifts among graphs. To solve these challenges and achieve better generalizability, we innovatively propose a **C**onstrained **V**ariational **G**eneration (CVG) framework to generate generalizable graphs for classification. Our framework is implemented based on the Variation Graph Auto-Encoder (VGAE) structure and optimized under the guidance of the Graph Information Bottleneck (GIB) principle, with its effectiveness validated by our theoretical analysis. We conduct extensive experiments on real-world datasets and demonstrate the superiority of our framework over state-of-the-art baselines.

## 1 INTRODUCTION

Graph-structured data is present in various crucial real-world domains, such as social networks, biological networks, and chemical molecules (Xu et al., 2019). Recently, relevant works have focused on learning graph representations by encoding graphs into vectorized representations (Wu et al., 2019). Graph Neural Networks (GNNs) (Kipf & Welling, 2017; Veličković et al., 2018), for instance, employ an iterative learning mechanism to extract valuable graph information and have achieved success in various real-world applications. However, current methods for learning graph representations heavily rely on the assumption that testing and training graph data are independently drawn from the same distribution, known as the I.I.D. assumption. In real-world scenarios, distribution shifts of graph data can be prevalent and inevitable due to the uncontrollable underlying data generation processes (Wu et al., 2022a). Nonetheless, most existing approaches struggle to generalize well when faced with out-of-distribution (OOD) testing graph data (Gui et al., 2022).

More recently, various methods have been proposed to tackle the OOD generalization problem on graph data (Wu et al., 2022c; Chen et al., 2022; Li et al., 2022b). In particular, they primarily focus on exploiting the *invariant* property across different graph data distributions. More specifically, they extract invariant subgraphs from any given graph, such that the extracted subgraphs are generalizable and thus can be utilized as input for prediction (Li et al., 2022a; Buffelli et al., 2022). In general, they rely on the assumption that the invariant information in these extracted subgraphs is shared across various graph distributions and consequently more generalizable (Miao et al., 2022).

However, in practice, such an assumption can often be unrealistic, primarily due to two significant challenges, as illustrated in Fig. 1. (1) **Intra-Graph Correlations.** The target of extracting invariant subgraphs is to provide a domain-invariant subgraph for classification. However, such a strategy is suboptimal in entirely extracting invariant information, as the identification of distinct invariant subgraphs can be challenging due to the complex correlations present within various graphs (Bevilacqua et al., 2021). For example, due to the complicated interactions (as edges) of atoms (as nodes) in a molecule graph, extracting a specific node may inevitably include both invariant and variant information, thus reducing the effectiveness of the extracted invariant subgraph (Gui et al., 2022). As a result, the strategy of extracting subgraphs struggles to maximally incorporate invariant information. (2) **Inter-Graph Distinctions.** The strategy of extracting invariant subgraphs faces substantial lim-

itations in the presence of significant distribution shift (Zhang et al., 2021; Li et al., 2022a), where the invariant subgraphs are less informative or even do not exist. For example, if an unseen test molecule largely deviates from the training molecules in terms of graph structures, the model will easily fail to extract invariant information that is shared across training and test molecules (Li et al., 2022a). Therefore, the resulting subgraph can be uninformative and unhelpful for classification.

To address the above two challenges, we innovatively propose a generative framework that aims to tackle the graph OOD generalization problem via **C**onstrained **V**ariational **G**eneration (CVG). In particular, our framework consists of two crucial strategies to solve the two challenges, respectively: (1) *Varaitional Generation*. Instead of extracting invariant subgraphs, we propose to *generate* entirely new graphs for classification. This design allows us to flexibly preserve valuable information without the need for extracting discrete structures, thereby ensuring that the intra-graph correlation information can be maximally preserved. The incorporation of variation also strengthens the robustness of the generation (Miao et al., 2022). (2) *Optimization Constraints*. To effectively handle potentially significant distribution shifts, we propose to leverage the Graph Infomation Bottleneck (GIB) principle (Yu et al., 2021) to *constrain* the optimization of the generator. Based on our theoretical analysis of GIB, we further introduce a regularization loss and a generation loss that can reduce the reliance of the generator on the original graph while focusing on label-relevant information. In consequence, the generated graphs can be more generalizable to handle distinct distribution shifts. In summary, our contributions are as follows:

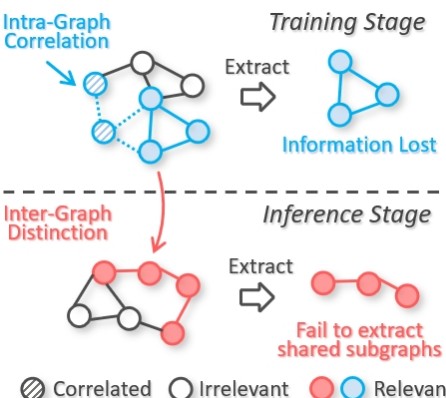

Figure 1: The two challenges of existing works that extract invariant subgraphs.

- **Problem**. We investigate the challenges of extracting invariant subgraphs in OOD generalization on graphs: from the perspectives of *Intra-Graph Correlations* and *Inter-Graph Distinctions*. We further discuss the necessity of tackling these challenges to enhance model generalizability.

- **Method**. We develop a novel framework with two essential strategies to learn generalizable graph representations: (1) a variational generative framework to generate graphs that can maximally leverage correlation information for generalizability; (2) a GIB-based strategy that constrains the optimization of the generator to handle inter-graph distinctions based on our theoretical analysis.

- **Experiments**. We conduct extensive experiments on four graph datasets that cover both synthetic and real-world out-of-distribution data. The results of significant performance improvement further demonstrate the superiority of our proposed framework over other state-of-the-art baselines.

## 2 PROBLEM FORMULATION

In this section, we present the formulation for the out-of-distribution (OOD) generalization problem on graphs. Generally, we are provided with a set of graph datasets $\mathcal{D}_{all} = \{\mathcal{D}_1, \mathcal{D}_2, \ldots, \mathcal{D}_{|\mathcal{D}_{all}|}\}$, referred to as different environments (Chen et al., 2022). Sampling $(G, Y) \sim P(G, Y|\mathcal{D}_i)$ from a specific environment $\mathcal{D}_i$ can be considered as drawn independently from an identical distribution $P(G, Y|\mathcal{D}_i)$ (Zhu et al., 2021; Zhang et al., 2022), where $G \in \mathbb{G}$, and $Y \in \mathbb{Y}$ represents the associated label. Here $\mathbb{G}$ and $\mathbb{Y}$ denote the graph space and the label space, respectively. Moreover, we can represent a graph as $G = (\mathcal{V}, \mathcal{E}, X)$ with its associated label $Y$, where $G \in \mathbb{G}$ and $Y \in \mathbb{Y}$. $\mathcal{V}$ and $\mathcal{E}$ denote the sets of nodes and edges, respectively. Additionally, $X = \{\mathbf{x}_i\}_{i=1}^{|\mathcal{V}|}$ represents the set of node features, where $\mathbf{x}_i$ is the $d_x$-dimensional attributes of the $i$-th node.

It is noteworthy that in graph datasets, the environment label for graphs is *unobserved*, as it is excessively expensive and laborious to assign environment labels for graphs in real-world applications (Li et al., 2022b; Wu et al., 2022c;b). Therefore, the overall graph data can be split into the training environment $\mathcal{D}_{tr}$ and the test environment, i.e., $\mathcal{D}_{all} = \{\mathcal{D}_{tr}, \mathcal{D}_{te}\}$. We can denote the graphs in each environment as $\mathcal{D}_{tr} = \{(G_1, Y_1), (G_2, Y_2), \ldots, (G_{|\mathcal{D}_{tr}|}, Y_{|\mathcal{D}_{tr}|})\}$ and $\mathcal{D}_{te} = \{(G_1, Y_1), (G_2, Y_2), \ldots, (G_{|\mathcal{D}_{te}|}, Y_{|\mathcal{D}_{te}|})\}$.

Following existing works, the overall process of OOD graph generalization can be formulated via a classification function $f(\cdot)$ and a mapping function $g(\cdot)$. Generally, $g(\cdot)\colon \mathbb{G} \to \mathbb{G}$ maps a raw input graph to another (invariant) graph for classification. Note that in previous works, $g(\cdot)$ is typically defined as an invariant identifier to extract subgraphs, while in our framework, we consider $g(\cdot)$ as a generator to output generalizable graphs $C$. $f(\cdot)\colon \mathbb{G} \to \mathbb{Y}$ is the classifier that takes the graph obtained from $g(\cdot)$ as input and aims to predict its label $Y$. By introducing a loss function $\ell$, we can express the empirical risk (i.e., the optimization objective) of classifier $f(\cdot)$ and generator $g(\cdot)$ under environment $\mathcal{D}_{tr}$ as follows:

$$\mathcal{R}(f, g, \mathcal{D}_{tr}) = \mathbb{E}_{(G,Y)\sim\mathbb{P}(G,Y|\mathcal{D}_{tr})}[\ell\left(f\left(g(G; \mathcal{D}_{tr})\right), Y\right)]. \qquad (1)$$

By optimizing the above risk, we aim to train a classifier $f(\cdot)$ and a generator $g(\cdot)$ that minimize the risk on the test environment, i.e., $\mathcal{R}(f, g, \mathcal{D}_{te})$.

## 3 METHODOLOGY

In this section, we elaborate on our proposed framework CVG, which aims to tackle the OOD graph generalization problem via constrained variational generation of generalizable graphs.

### 3.1 CONSTRAINED VARIATIONAL GENERATION (CVG) FOR OOD GENERALIZATION

We innovatively propose to tackle the OOD generalization problem via Constrained Variational Generation (CVG). We specify our goal as generating generalizable graphs via a generator trained in constraints such as class labels. It is noteworthy that CVG only constrains the generation during training, unlike conditional generation methods that incorporate specific information during both training and inference (Sohn et al., 2015; Mishra et al., 2018; Kim et al., 2021). This is because we have no access to the label during inference, and thus it is unreasonable and impractical to condition on labels. We first formulate the objective of CVG as follows,

$$\max P(C|G), \quad \text{s.t. } \mathrm{KL}\left(P(Y|G)\|P(Y|C)\right) \leq \gamma, \qquad (2)$$

where $\gamma \geq 0$ is used to constrain the KL-divergence between the $P(Y|G)$ and $P(Y|C)$, i.e., the label distributions of the original graph $G$ and the generalizable graph $C$, respectively.

Notably, although we propose the CVG objective in Eq. (2), it is difficult to optimize as the result can easily lead to a trivial solution where $C = G$ for any given $G$. Therefore, we propose to optimize it via the Graph Information Bottleneck principle.

### 3.2 GRAPH INFORMATION BOTTLENECK FOR CVG

**GIB Objective.** The idea of Graph Information Bottleneck (GIB) is prevalently used to guide the learning of label-relevant parts such as subgraphs and promote model generalizability by reducing the effect of the label-irrelevant part (Tishby & Zaslavsky, 2015; Wu et al., 2020; Yu et al., 2021). To leverage the GIB principle for our framework, we formulate the GIB objective as follows:

$$\max \ I(C; Y) - \beta I(C; G), \qquad (3)$$

where $\beta \geq 0$ is a scalar to control the weight of the first term, and $C$ is the generalizable graph we aim to generate. The main idea of GIB is to learn specific representations for graphs such that they maintain the crucial label-relevant information for classification while reducing the mutual information between them and the original graphs.

**Latent Representation $Z$.** Although GIB is proven to be effective (Miao et al., 2022) in OOD generalization, it is difficult to leverage it to generate $C$ from $G$ based on the simple objective. Therefore, we leverage a variational latent representation $Z$ that is directly related to $C$ and $G$, as illustrated in Fig. 2. Based on the chain rule, we can transform the second term as $I(C; G) = I(C; G, Z) - I(C; Z|G)$. In this manner, the objective of generating $C$ with $Z$ is as follows:

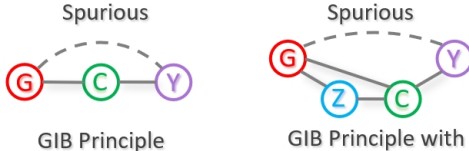

Figure 2: The illustration of the original GIB and the GIB with latent $Z$.

$$I(C; Y) - \beta I(C; G) = I(C; Y) - \beta I(C; G, Z) + \beta I(C; Z|G). \qquad (4)$$

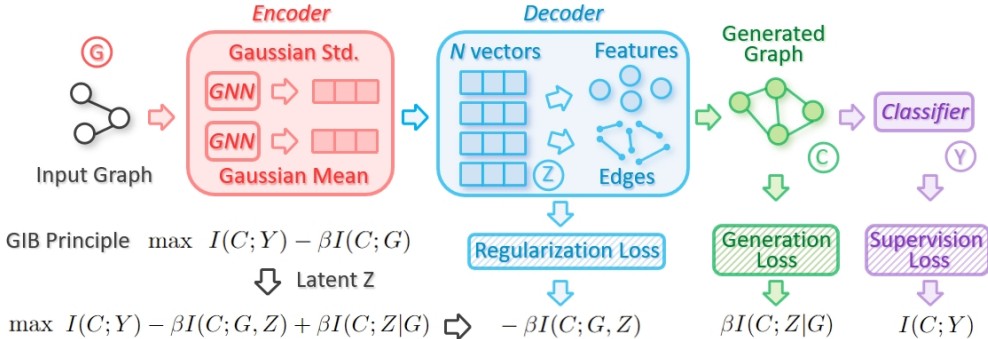

Figure 3: The overall framework of CVG. For each input graph, we leverage a VGAE, comprised of an encoder and decoder, to process it into a Gaussian distribution and sample $N$ times from it to generate a new graph $C$. Then $C$ is input into the classifier to obtain prediction results. We optimize the framework based on the GIB principle, and each of the three terms is transformed into a loss.

By decomposing the term $I(C; G)$ into two terms involving the latent representations $Z$, we construct a connection between the generalizable graph $C$ and the original $G$ via $Z$. Such a strategy enables the utilization of variational inference to learn more generalizable $Z$ and $C$.

**Lower Bound.** Nevertheless, the objective in Eq. (4) is still intractable. To enable the optimization in a parameterized manner, we derive a lower bound for each term in it to enable the optimization of the objective. The detailed proof of the following theorem is provided in Appendix A.

**Theorem 3.1.** *Introducing a variational approximation Q(C), a lower bound for the GIB objective is as follows:*

$$
\begin{aligned}
I(C;Y) - \beta I(C;G) \geq{} & \mathbb{E}\left[\log P(Y|G)\right] + H(Y) - KL\left(P(Y|G)\|P(Y|C)\right) \\
& - \beta\mathbb{E}[KL(P_g(C|G,Z)\|Q(C))] + \beta\mathbb{E}\left[\log\left(P_g(C|G,Z)\right)\right] + \beta H(C|G).
\end{aligned}
\tag{5}
$$

The notation $KL(\cdot\|\cdot)$ represents the Kullback-Leibler (KL) divergence. Naturally, the term $P(Y|C)$ and $P(C|G,Z)$ can be considered as the classifier $f(\cdot)\colon \mathbb{G} \to \mathbb{Y}$ and the generator $g(\cdot)\colon \mathbb{G} \to \mathbb{G}$, respectively, as defined in the problem formulation in Sec. 2.

We consider $P(Y|G)$ as a deterministic distribution for each $G$, and thus it can be neglected during optimization. Similarly, $H(Y)$ and $H(C|G)$ can also be considered as a constant. Therefore, by rearranging Eq. (5), we can achieve the final optimization objective for GIB in Eq. (3) as follows:

$$
\max\ \beta\mathbb{E}\left[\log\left(P_g(C|G,Z)\right)\right] - \beta\mathbb{E}[KL(P_g(C|G,Z)\|Q(C))] - KL\left(P(Y|G)\|P(Y|C)\right).
\tag{6}
$$

### 3.3 Constrained Variational Generation via VGAE

Although we have derived the objective for optimization with GIB, it remains challenging to model $Z$ with suitable generator architectures, especially in the absence of the ground truth $C$. Particularly, we propose to leverage the VGAE structure (Kipf & Welling, 2016; Simonovsky & Komodakis, 2018) for generation. The reason is that the VGAE optimization objective, under the constraints based on class labels, aligns with the GIB objective according to our derivation presented below. The detailed proof is provided in Appendix B.

**Theorem 3.2.** *By deriving the evidence lower bound (ELBO) for $P(C|G)$, the VGAE objective for constrained variational generation (CVG), with a hyper-parameter $\lambda$, is as follows:*

$$
\max\ \mathbb{E}_Q[\log P(C|G,Z)] - KL(Q(Z)\|P(Z|G)) - \lambda KL\left(P(Y|G)\|P(Y|C)\right).
\tag{7}
$$

In Theorem 3.2, we theoretically validate that the constrained generation objective for VGAE in Eq. (7) complies with the GIB objective in Eq. (6) with $\beta = 1/\lambda$, except for the KL-divergence $KL(Q(Z)\|P(Z|G)$ that acts as a regularization term. Therefore, leveraging the VGAE structure as our generator can ease the difficulty of optimizing our framework based on the GIB principle.

In Fig. 3, we present the architecture of our generator, which consists of an encoder and a decoder, following the conventional structure of VAEs (Kingma et al., 2019). The encoder, implemented as

a GNN, processes the input graph $G = (\mathcal{V}, \mathcal{E}, X)$ and then computes a stochastic latent variable $Z = \{\mathbf{z}_i\}_{i=1}^{N}$. Here $\mathbf{z}_i \in \mathbb{R}^{d_z}$ corresponds to the latent variable of a specific node (Kipf & Welling, 2016; Yang et al., 2019). Here $N$ is the number of nodes in each generated graph, which is a controllable hyper-parameter. Notably, to keep the consistency of generated graphs, we use the same value of $N$ throughout training and inference, with its benefit evaluated in Sec. 4.3. The decoder then utilizes $Z$ to generate the generalizable graph $C$. Specifically, we generate $Z = \{\mathbf{z}_i\}_{i=1}^{N}$ by individually sampling $N$ times from a single distribution $\mathcal{N}(\mathbf{z}|\mu, \mathrm{diag}(\sigma^2))$ as follows:

$$\mathbf{z}_i \sim \mathcal{N}(\mathbf{z}|\mu, \mathrm{diag}(\sigma^2)), \; i = 1, 2, \ldots, N,$$

$$\text{where } \mu = \frac{1}{|\mathcal{V}|} \sum_{j=1}^{|\mathcal{V}|} \mathrm{GNN}_\mu(\mathcal{V}, \mathcal{E}, X)_j, \text{ and } \log \sigma = \frac{1}{|\mathcal{V}|} \sum_{j=1}^{|\mathcal{V}|} \mathrm{GNN}_\sigma(\mathcal{V}, \mathcal{E}, X)_j. \tag{8}$$

Here $\mathrm{GNN}_\mu$ and $\mathrm{GNN}_\sigma$ are two separate GNNs used for generating the mean value and standard deviation for sampling the latent variable $\mathbf{z}_i$, respectively. Moreover, $\mathrm{GNN}(\mathcal{V}, \mathcal{E}, X)_i$ denotes the $i$-th row vector of $\mathrm{GNN}(\mathcal{V}, \mathcal{E}, X)$. In addition, we calculate the logarithmic value of the standard deviation (i.e., $\log \sigma$) instead of direct computation of $\sigma$, which is to smoothly scale the deviation and enable the appearance of smaller or larger values for $\sigma$.

In the above process, the learning of $\mathbf{z}_i$ can be considered as sampling from the average of all learned latent variables on $G$, thus being a graph embedding of $G$. In particular, the reasons for learning a single latent distribution instead of multiple ones are two-fold: (1) We can control the number of nodes in the generated consistent graphs. In works (Kipf & Welling, 2016; Simonovsky & Komodakis, 2018) that learn an individual latent distribution for each node, the resulting generated graph has the same number of nodes with $\mathcal{V}$, which is rather inflexible. (2) More importantly, this strategy aligns with the KL-divergence term $\mathbb{E}[\mathrm{KL}(P_g(C|G, Z)\|Q(C))]$. This term implies that the generated classification $C$ should be close to a prior distribution. Therefore, we fix the number of generated nods across training and inference to better minimize the KL-divergence.

With the obtained latent representations $Z$, we can generate node features $X^*$ as follows:

$$X^* = \{\mathbf{x}_1^*, \mathbf{x}_2^*, \ldots, \mathbf{x}_N^*\}, \text{ where } \mathbf{x}_i^* = \mathbf{W}_i \mathbf{z}_i. \tag{9}$$

Here $\mathbf{W}_i \in \mathbb{R}^{d_x^* \times d_z}$ is the projection layer weight for $\mathbf{z}_i$. $d_x^*$ and $d_z$ are the dimension sizes for $X^*$ and $Z$, respectively. Then we further generate edges from the latent variables $\mathbf{z}_i$ as follows:

$$P(\mathcal{E}^*) = \prod_{i=1}^{N} \prod_{j=1}^{N} p(e_{ij}^*|\mathbf{z}_i, \mathbf{z}_j), \text{ where } p(e_{ij}^* = 1|\mathbf{z}_i, \mathbf{z}_j) = \alpha_{ij} = \sigma((\mathbf{W}_i \mathbf{z}_i)^\top \cdot \mathbf{W}_j \mathbf{z}_j). \tag{10}$$

Here $e_{ij}^* = 1$ if there exists an edge between the $i$-th node and the $j$-the node in the generated graph, and $e_{ij}^* = 0$, otherwise. That being said, each $e_{ij}^* \sim \mathrm{Bernoulli}(\alpha_{ij})$ follows a Bernoulli distribution. $\sigma(x) = 1/(1 + \exp(-x))$ is the Sigmoid function. Note that directly sampling edges based on Eq. (10) cannot provide computable gradients. Therefore, to ensure optimization based on gradient descent, we adopt the gumbel-sigmoid (Jang et al., 2017) strategy to sample edges. In this way, we can generate a new generalizable graph $C$ based on Eq. (8), Eq. (9), and Eq. (10):

$$C = (\mathcal{V}^*, \mathcal{E}^*, X^*) = g(G), \text{ where } G = (\mathcal{V}, \mathcal{E}, X). \tag{11}$$

It is noteworthy that we stochastically generate discrete edges, instead of using continuous edge weights. Such a strategy provides an additional way to extract useful information from the generated $C$, thus enhancing the generalizability of $C$. Its effectiveness is empirically verified in Sec. 4.3. Note that during inference, we remove all stochasticity and only keep the top-ranked edges.

### 3.4 Optimization based on GIB

In this subsection, we introduce how to optimize our framework according to GIB the objective described in Eq. (6). We formulate the three terms into three different losses, respectively.

**Supervision Loss.** For the term $-\beta \mathrm{KL}(P(Y|G)\|P(Y|C))$, maximizing it is equal to minimizing the discrepancy between the predicted labels of generated $C$ and the original graph $G$. Specifically, we consider the value of $p(y|G)$ as 1 if $y$ is the label of $G$, and $P(y|G) = 0$, otherwise. In this manner, the loss for maximizing this term is formulated as follows:

$$\mathcal{L}_s(C, G, Y) = - \sum_{y \in \mathbb{Y}} p(y|G) \log p(y|C). \tag{12}$$

In practice, the classification probabilities (i.e., $p(y|C)$) are achieved by inputting $C$ into the classifier $f(\cdot)$, which is implemented as a GNN and an MLP.

**Regularization Loss.** Considering the term $-\mathbb{E}[\text{KL}(P_g(C|G, Z)\|Q(C))]$, maximizing it is equal to reducing the difference between the distribution of $C$ (given $G$ and $Z$) and the prior distribution $Q(C)$. Notably, the lower bound derived in Eq. (5) is true for any $Q(C)$. Particularly, we define $Q(C)$ as follows. As the generated $C$ consists of two components, i.e., node features $X^*$ and edges $\mathcal{E}$, we decompose $Q(C)$ into two parts: $Q(C) = Q(X, \mathcal{E}) = Q(X) \cdot Q(\mathcal{E})$. As a result, the original term becomes $-\mathbb{E}[\text{KL}(P_g(X|G, Z)\|Q(X)) + \text{KL}(P_g(\mathcal{E}|G, Z)\|Q(\mathcal{E}))]$, as the generation of $X$ and $\mathcal{E}$ are inherently decoupled in our framework. In practice, we define the prior distribution $Q(X)$ as a Gaussian distribution, i.e., $\mathcal{N}(0, \mathbf{I})$, where $\mathbf{I} \in \mathbb{R}^{d_x^* \times d_x^*}$ is an identity matrix. For the second KL-divergence term, we formulate $Q(\mathcal{E})$ as $|\mathcal{E}|$ independent and identical Bernoulli distributions: $e_{ij}^* \sim \text{Bernoulli}(\epsilon)$, where $\epsilon \in [0, 1]$ is a pre-defined hyper-parameter. In this manner, we can consider the generation of the edge between any node pair in $C$ as a Bernoulli distribution and calculate the KL-divergence between them and $Q(\mathcal{E})$. Note that as $X$ is linearly projected from $Z$, we use the values of $Z$ for calculation. In summary, the regularization loss, consisting of two KL-divergence terms, can be formulated as follows:

$$\mathcal{L}_r(C) = \frac{1}{2}\sum_{i=1}^{d_x^*}(\sigma_i^2 + \mu_i^2 - 2\log\sigma_i) + \sum_{i=1}^{N}\sum_{j=i}^{N}\beta_{ij}\log\frac{\alpha_{ij}}{\epsilon} + (1 - \alpha_{ij})\log\frac{1 - \alpha_{ij}}{1 - \epsilon}. \quad (13)$$

The regularization term on the latent variables and the generated edges can ensure that the model behaves in a more predictable manner, thereby potentially improving the generalizability even in the absence of the ground truth $C$.

**Generation Loss.** For the last term $\mathbb{E}[\log(P_g(C|G, Z))]$, the direct optimization is infeasible, as we lack the ground truth, i.e., the optimal $C$ for each $G$. Therefore, we propose a similarity loss base on the intuition that the generated $C$ should be similar to the same $Z$ while different from $C$ generated from other $G$.

$$\mathcal{L}_g(C) = -\sum_{i=1}^{S}\text{sim}(C, C_i)/\tau + \log\left(\sum_{k=1}^{K}\exp\left(\text{sim}(C, C_k')/\tau\right)\right), \quad (14)$$

where $S$ and $K$ are the numbers of positive samples and negative samples, respectively. $\tau \geq 0$ is the temperature parameter. Specifically, we sample positive samples by repeatedly generating multiple $C$ from $G$. For negative samples, we choose $C$ with different labels from the positive samples. The similarity function $\text{sim}(C, C_i)$ is implemented as the dot product of the hidden representations of $C$ and $C_i$ learned by $f(\cdot)$ before the final MLP layer.

In summary, the supervision loss $\mathcal{L}_s$ acts as a term that helps our framework learn from the class labels of graphs, which are the mere supervision information in OOD generalization on graphs. The regularization loss $\mathcal{L}_r$ is used to constrain the generated graphs $C$ in a predictable way, such that they can be more generalizable to unseen distributions. The similarity loss aims to provide a more stable generation of $C$ and more clearly distinguish $C$ of different classes. Combining these three losses, we can formulate the training objective of our framework CVG as follows:

$$\mathcal{L} = \mathbb{E}_{(G,Y)\sim\mathbb{P}(G,Y|\mathcal{D}_{tr})}\left[\mathcal{L}_s(C, G, Y) + \beta_r\mathcal{L}_r(C) + \beta_g\mathcal{L}_g(C)\right], \text{ where } C = g(G). \quad (15)$$

Here $\beta_r$ and $\beta_g$ are two hyper-parameters that control the importance of $\mathcal{L}_r$ and $\mathcal{L}_g$, respectively. With the proposed loss, we can effectively optimize the objective for the constrained variational generation to tackle the OOD generalization problem on graph data.

### 3.5 DISCUSSION

The major difference between CVG and existing works (also the main contribution of our work) is to leverage a variational generative framework to tackle the OOD generalization problem on graphs. To the best of our knowledge, CVG is the first to propose a generative workflow for learning generalizable graph representation. Although multiple works (Guo et al., 2020; Liu et al., 2022) also use generators for OOD generlization, they do not directly use generated graphs for predictions. Moreover, by introducing variational generation, the generated graphs are also more generalizable as the learned information on them is well preserved.

Table 1: The graph OOD generalization results (Test accuracy in % for SP-Motif, MNIST-75sp, and Graph-SST2, ROC-AUC for Molhiv). The best results are in **bold**.

| Dataset | SP-Motif | | | | MNIST-75sp | Graph-SST2 | Molhiv |
|---------|----------|--------|--------|--------|------------|------------|--------|
| | *Balanced* | $b = 0.5$ | $b = 0.7$ | $b = 0.9$ | | | |
| ERM | $42.99_{\pm 1.93}$ | $39.69_{\pm 1.73}$ | $38.93_{\pm 1.74}$ | $33.61_{\pm 1.02}$ | $12.71_{\pm 1.43}$ | $81.44_{\pm 0.59}$ | $76.20_{\pm 1.14}$ |
| Attention | $43.07_{\pm 2.55}$ | $39.42_{\pm 1.50}$ | $37.41_{\pm 0.86}$ | $33.46_{\pm 0.43}$ | $15.19_{\pm 2.62}$ | $81.57_{\pm 0.71}$ | $75.84_{\pm 1.33}$ |
| ASAP | $44.44_{\pm 8.19}$ | $44.25_{\pm 6.87}$ | $39.19_{\pm 4.39}$ | $31.76_{\pm 2.89}$ | $15.54_{\pm 1.87}$ | $81.57_{\pm 0.84}$ | $73.81_{\pm 1.17}$ |
| Top-k Pool | $43.43_{\pm 8.79}$ | $41.21_{\pm 7.05}$ | $40.27_{\pm 7.12}$ | $33.60_{\pm 0.91}$ | $14.91_{\pm 3.25}$ | $79.78_{\pm 1.35}$ | $73.01_{\pm 1.65}$ |
| GIB | $41.52_{\pm 1.22}$ | $36.09_{\pm 1.61}$ | $35.15_{\pm 2.05}$ | $33.94_{\pm 2.15}$ | $15.17_{\pm 1.38}$ | $80.14_{\pm 1.79}$ | $76.12_{\pm 2.64}$ |
| GSN | $43.18_{\pm 5.65}$ | $34.67_{\pm 1.21}$ | $34.03_{\pm 1.69}$ | $32.60_{\pm 1.75}$ | $19.03_{\pm 2.39}$ | $82.54_{\pm 1.16}$ | $74.53_{\pm 1.90}$ |
| GSAT | $74.95_{\pm 2.18}$ | $69.72_{\pm 1.93}$ | $67.31_{\pm 1.86}$ | $61.49_{\pm 3.46}$ | $24.93_{\pm 1.30}$ | $82.81_{\pm 0.56}$ | $80.67_{\pm 0.95}$ |
| IRM | $42.26_{\pm 2.69}$ | $41.30_{\pm 1.28}$ | $40.16_{\pm 1.74}$ | $35.12_{\pm 2.71}$ | $18.62_{\pm 1.22}$ | $81.01_{\pm 1.13}$ | $74.46_{\pm 2.74}$ |
| Group DRO | $41.51_{\pm 1.11}$ | $39.38_{\pm 0.93}$ | $39.32_{\pm 2.23}$ | $33.90_{\pm 0.52}$ | $15.13_{\pm 2.83}$ | $81.29_{\pm 1.44}$ | $75.44_{\pm 2.70}$ |
| V-REx | $42.83_{\pm 1.59}$ | $39.43_{\pm 2.69}$ | $39.08_{\pm 1.56}$ | $34.81_{\pm 2.04}$ | $18.92_{\pm 1.41}$ | $81.76_{\pm 0.08}$ | $75.62_{\pm 0.79}$ |
| DIR | $47.03_{\pm 2.46}$ | $45.50_{\pm 2.15}$ | $43.36_{\pm 1.64}$ | $39.87_{\pm 0.56}$ | $20.36_{\pm 1.78}$ | $83.29_{\pm 0.53}$ | $77.05_{\pm 0.57}$ |
| GIL | $55.44_{\pm 3.11}$ | $54.56_{\pm 3.02}$ | $53.12_{\pm 2.18}$ | $46.04_{\pm 3.51}$ | $21.94_{\pm 0.38}$ | $83.44_{\pm 0.37}$ | $79.08_{\pm 0.54}$ |
| CIGA | $76.52_{\pm 4.39}$ | $71.58_{\pm 3.57}$ | $68.25_{\pm 4.70}$ | $64.01_{\pm 3.17}$ | $25.29_{\pm 2.53}$ | $81.02_{\pm 1.29}$ | $79.75_{\pm 1.06}$ |
| CVG (Ours) | $\mathbf{79.62}_{\pm 2.24}$ | $\mathbf{76.57}_{\pm 2.89}$ | $\mathbf{72.25}_{\pm 3.12}$ | $\mathbf{65.80}_{\pm 2.95}$ | $\mathbf{30.12}_{\pm 1.25}$ | $\mathbf{84.21}_{\pm 0.99}$ | $\mathbf{81.09}_{\pm 0.72}$ |

Moreover, our framework is also capable of using different generative architectures, as another contribution of our work is the derivation of the GIB principle for optimization. Compared to GSAT (Miao et al., 2022) that stochastically learn edge weights, CVG preserves higher flexibility as we also generate node features and different graph structures. Furthermore, we provide theoretical justification for the GIB-based objective that involves a latent variable $Z$. In this manner, we can effectively train CVG without the ground truth of $C$.

## 4 EXPERIMENTS

### 4.1 EXPERIMENTAL SETTINGS

**Datasets.** We utilize a combination of synthetic and real datasets for OOD generalization tasks. Further details on dataset statistics, GNN architectures, and the training process are in Appendix C.

- **SP-Motif** (Ying et al., 2019): This is a synthetic dataset, where each graph is composed of a base (Tree, Ladder, and Wheel) and a motif (Cycle, House, and Crane). The graph label is solely determined by the motif. The false relations between the base and the label are introduced and controlled by a hyper-parameter $b$. The distribution shifts are from the inclusion of spuiors bases.

- **MNIST-75sp** (Knyazev et al., 2019): This dataset converts each MNIST image into a superpixel graph of ten classes. Superpixels represent nodes, while edges denote the distance between nodes. The distribution shifts are created by adding random noises.

- **Graph-SST2** (Socher et al., 2013; Yuan et al., 2022): Graph-SST2 comprises graphs labeled according to sentence sentiment, where nodes represent tokens, and edges indicate node relations. The distribution shifts are based on average node degrees.

- **Molhiv** (OGBG-Molhiv) (Wu et al., 2018; Hu et al., 2020; 2021): This molecule graph dataset is designed for molecular property prediction, where nodes represent atoms, and edges are chemical bonds. Each graph is labeled based on whether a molecule inhibits HIV replication or not. The distribution shifts originate from the inhere distinctions in various molecular structures.

**Baselines.** To provide a fair and thorough evaluation of our framework, we conduct experiments on two groups of baselines. (1) **Interpretable GNNs.** This group consists of methods proposed for explainability on graphs, including Graph Attention Networks (Veličković et al., 2018), Top-$k$ Pool (Gao & Ji, 2019), ASAP (Ranjan et al., 2020), GIB (Yu et al., 2021) , GSN (Bouritsas et al., 2022), and GSAT (Miao et al., 2022). (2) **Invariant Learning Methods.** This group includes baselines proposed for out-of-distribution generalization, including IRM (Arjovsky et al., 2019), Group

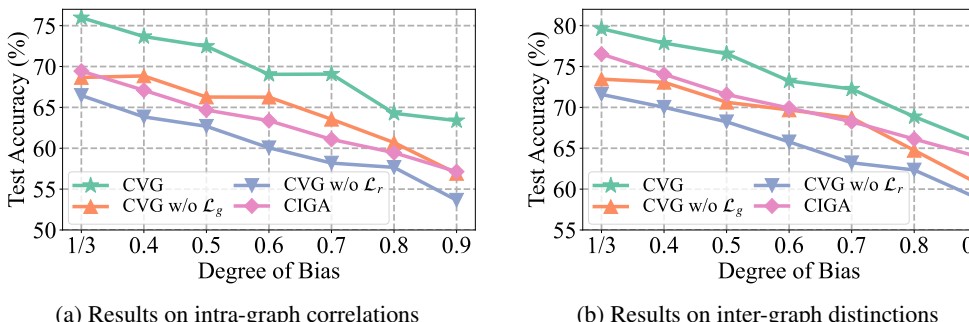

(a) Results on intra-graph correlations     (b) Results on inter-graph distinctions

Figure 4: The results on varying degrees of intra-graph correlations and inter-graph distinctions, obtained on SP-Motif-Cor and SP-Motif, respectively. Here $b = 1/3$ denotes the balanced setting.

DRO (Sagawa et al., 2020), V-REx (Krueger et al., 2021), DIR (Wu et al., 2022c), GIL (Li et al., 2022b), and CIGA (Chen et al., 2022). For each dataset, we utilize the same GNN structure for all baselines as well as our framework CVG. More details of the baselines are provided in Appendix C.

**Evaluation.** For the evaluation metrics, we use ROC-AUC for Molhiv, and Accuracy for other datasets. We run the experiments with ten different seeds and calculate the average and standard deviation for the results of each method and dataset.

### 4.2 Comparative Results

In this subsection, we evaluate our framework CVG and all baselines on the four datasets, with details provided in Appendix C. From the results in Table 1, we can obtain the following observations:

- **CVG consistently outperforms other baselines on all datasets.** Specifically, CVG archives a significant performance improvement on dataset SP-Motif and MNIST-75sp, compared to other state-of-the-art baselines. Although the types of major distribution shifts are different (structural shift for SP-Motif and feature shift for MNIST-75sp), CVG still outperforms other baselines on these two datasets. On other datasets, CVG also achieves better performance with generally lower variance. The overall results strongly validate that CVG preserves better generalization ability under different environments.

- **CVG preserves generalizability even with a large bias degree.** From the results on dataset SP-Motif with different degrees of bias, we can observe that interpretable GNN methods generally encounter a significant performance drop when the bias degree increases. This is primarily because their strategy of extracting subgraphs cannot effectively tackle the problem of inter-graph distinctions, thus leading to suboptimal performance. Nevertheless, our framework CVG consistently outperforms all baselines, even in the presence of a large bias.

### 4.3 Ablation Study

**Effects of Graph Structures on Intra-Graph Correlations.** We evaluate the effectiveness of our framework in terms of tackling the challenge of intra-graph correlations. As it is difficult to measure the degree of such correlations in real-world datasets, we create a novel synthetic dataset, SP-Motif-Cor, with node features correlated across graph structures (details provided in Appendix C). The results are presented in Fig. 4a. We first observe that CVG consistently outperforms the strongest baseline CIGA and other variants under all bias degrees, highlighting that our method can effectively handle the problem of intra-graph correlations. The results further demonstrate that the generation of graphs can tackle such challenges when the degree is higher.

**Effects of GIB-based Optimization on Inter-Graph Distinctions.** We evaluate the robustness of our framework under various degrees of inter-graph distinctions. We utilize the dataset SP-Motif and present the results in Fig. 4b. Specifically, we can observe that CVG is robust to different degrees of inter-graph distinctions and achieves better results than other varaints. Moreover, the superiority of CVG over CIGA and the variant without regularization loss reflects the effects of leveraging the GIB principle for optimization.

**Effects of Regularization Loss and Graph Size.** We aim to explore the joint effects of the regularization loss weight $\beta_r$ and the number of nodes $N$ in the generated graph $C$. We present the results on SP-Motif with $b = 0.5$ in Fig. 5. From the results, we can observe that, in general, increasing the weight of the regularization loss will bring performance improvement. However, an excessively large value of $\beta_r$ will adversely impact the performance. Similarly, the number of nodes around $N = 5$ is more suitable for OOD generalization on SP-Motif. Moreover, we observe that with a larger $N$, the optimal weight of the regularization loss also increases. This observation demonstrates that a larger generated graph requires more regularization to ensure its stability.

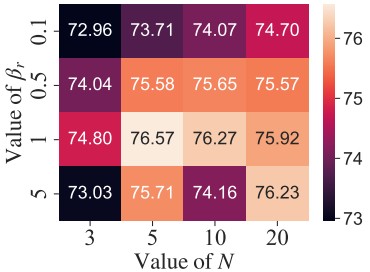

Figure 5: The results of regularization loss and the number of nodes

**Effects of Variational Generation.** In Fig. 6, we present the results on SP-Motif using static generation, i.e., removing the sampling of generated node features and edges, denoted as CVG w/o V. Here, the edges are directly assigned the continuous values of $\alpha_{ij}$. We also include the results without the regularization loss (CVG w/o R) to explore their effect on both static and variation generation. We use CVG w/o VR to denote the variant that removes both. From the results, we can observe that for CVG w/o V, the performance drops the most, as the learned representations are less informative. Without regularization, the static generation performs the second worst. Moreover, variation generation without

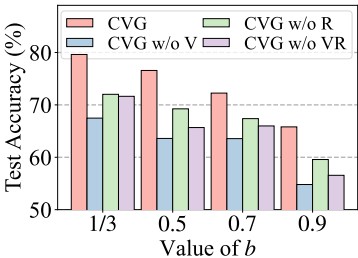

Figure 6: The results on variational generation and regularization.

regularization results in a larger variance in the generated $C$, thereby detrimenting the efficacy.

## 5 RELATED WORKS

**Out-of-Distribution Generalization.** Out-of-distribution (OOD) generalization pertains to the task of learning a model that is generalizable to unseen test distributions, trained on related yet distinct training data. Prior research on invariant learning (Arjovsky et al., 2019; Ganin & Lempitsky, 2015; Li et al., 2018a;b) typically focuses on establishing a consistent input-output relationship across different distributions, often by learning invariant features (Peng et al., 2019; Sun et al., 2016) or optimizing worst-case group performance (Hu et al., 2018; Sagawa et al., 2020). In contrast, adaptive methods for OOD generalization adapt the learned models to a specific domain (Kumagai & Iwata, 2018). For instance, ARM (Zhang et al., 2021) proposes an adaptive framework that extracts information from data points in the test domain for adaptation, while another work (Kumagai & Iwata, 2018) treats contexts from the test domain as probabilistic latent variables to achieve adaptation.

**Graph Out-of-Distribution Generalization.** Recently, there has been a growing interest in addressing the graph OOD generalization problem (Chen et al., 2022; Li et al., 2022b;a; Wu et al., 2022a;c; Miao et al., 2022). Among these approaches, DIR (Wu et al., 2022c) leverages a set of graph representations as causal rationales and performs interventional augmentations to generate additional distributions. GIL (Li et al., 2022b) and CIGA (Chen et al., 2022) both focus on learning invariant graph representations. While GIL aims to identify invariant subgraphs with a GNN, CIGA extracts subgraphs that optimally preserve invariant intra-class information. In contrast to these methods that extract invariant subgraphs, our framework resorts to generating new graphs for classification, which tackles the challenges of intra-graph correlations and inter-graph distinctions.

## 6 CONCLUSION

In this work, we investigate the crucial problem of out-of-distribution generalization on graphs. Specifically, we propose a novel framework for constrained variational generation, based on the VGAE structure and optimized according to the GIB principle. We provide further theoretical analysis to verify the effectiveness of our framework CVG, along with extensive experiments conducted on a variety of synthetic and real-world graph datasets. The results validate the superiority of our framework over other state-of-the-art baselines on graph out-of-distribution generalization tasks.

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

## A    THEOREM 3.1 AND PROOF

In this section, we provide proof for Theorem 3.1.

**Theorem 3.1.** *Introducing a variational approximation $Q(C)$, a lower bound for the GIB objective is as follows:*

$$\begin{aligned}
I(C;Y) - \beta I(C;G) \geq{}& \mathbb{E}\left[\log P(Y|G)\right] + H(Y) - KL\left(P(Y|G)\|P(Y|C)\right) \\
&- \beta\mathbb{E}[KL(P_g(C|G,Z)\|Q(C))] + \beta\mathbb{E}\left[\log\left(P_g(C|G,Z)\right)\right] + \beta H(C|G).
\end{aligned} \tag{16}$$

*Proof.* Specifically, we aim to transform or provide a lower bound for each of the three terms in

$$I(C;Y) - \beta I(C;G) = I(C;Y) - \beta I(C;G,Z) + \beta I(C;Z|G). \tag{17}$$

We provide the detailed deviation as follows:

$$\begin{aligned}
I(C;Y) &= \mathbb{E}_{C,Y}\left[\log\frac{P(Y|C)}{P(Y)}\right] \\
&= \mathbb{E}\left[\log\frac{P(Y|G)}{P(Y)}\right] - KL\left(P(Y|G)\|P(Y|C)\right) \\
&= \mathbb{E}\left[\log P(Y|G)\right] + H(Y) - KL\left(P(Y|G)\|P(Y|C)\right).
\end{aligned} \tag{18}$$

$$\begin{aligned}
I(C;Z|G) &= \mathbb{E}_{C,Z,G}\left[\log\frac{P(C,Z|G)}{P(C|G)P(Z|G)}\right] \\
&= \mathbb{E}_{C,Z,G}\left[\log\frac{P(C|G,Z)}{P(C|G)}\right] \\
&= \mathbb{E}_{C,Z,G}\left[\log\left(P(C|G,Z)\right)\right] + H(C|G)
\end{aligned} \tag{19}$$

For the following derivation, we introduce a variational approximation $Q(C)$:

$$\begin{aligned}
-I(C;G,Z) &= -\mathbb{E}_{C,G}\left[\log\left(\frac{P(C|G,Z)}{Q(C)}\right)\right] + KL(P(C)\|Q(C)) \\
&\geq -\mathbb{E}_{C,G}\left[\log\left(\frac{P(C|G,Z)}{Q(C)}\right)\right] \\
&= -\mathbb{E}_G[KL(P(C|G,Z)\|Q(C))].
\end{aligned} \tag{20}$$

Therefore, we can finally achieve the following results:

$$I(C;Y) = \mathbb{E}\left[\log P(Y|G)\right] + H(Y) - KL\left(P(Y|G)\|P(Y|C)\right) \tag{21}$$

$$I(C;G,Z) \leq \mathbb{E}_G[KL(P(C|G,Z)\|Q(C))]. \tag{22}$$

$$I(C;Z|G) = \mathbb{E}_{C,Z,G}\left[\log\left(P(C|G,Z)\right)\right] + H(C|G). \tag{23}$$

Combing the above three equations (or inequations), we can achieve the final results:

$$\begin{aligned}
I(C;Y) - \beta I(C;G) \geq{}& \mathbb{E}\left[\log P(Y|G)\right] + H(Y) - KL\left(P(Y|G)\|P(Y|C)\right) \\
&- \beta\mathbb{E}[KL(P_g(C|G,Z)\|Q(C))] + \beta\mathbb{E}\left[\log\left(P_g(C|G,Z)\right)\right] + \beta H(C|G).
\end{aligned} \tag{24}$$

$\square$

## B    THEOREM 3.2 AND PROOF

In this section, we provide proof for Theorem 3.2.

**Theorem 3.2.** *By deriving the evidence lower bound (ELBO) for $P(C|G)$, the VGAE objective for constrained variational generation (CVG), with a hyper-parameter $\lambda$, is as follows:*

$$\max\ \mathbb{E}_Q[\log P(C|G,Z)] - KL(Q(Z)\|P(Z|G)) - \lambda KL\left(P(Y|G)\|P(Y|C)\right). \tag{25}$$

*Proof.* We first derive an evidence lower bound (ELBO) for the VGAE objective $P(C|G)$:

$$
\begin{aligned}
&\log P(C|G) \\
&= \log \int_{\mathbf{Z}} P(C, \mathbf{Z}|G) d\mathbf{Z} \\
&= \log \int_{\mathbf{Z}} Q(\mathbf{Z}|G) \frac{P(C, \mathbf{Z}|G)}{Q(\mathbf{Z}|G)} d\mathbf{Z} \\
&\quad (\textit{using Jensen's Inequality}) \\
&\geq \int_{\mathbf{Z}} Q(\mathbf{Z}) \log \frac{P(C, \mathbf{Z}|G)}{Q(\mathbf{Z})} d\mathbf{Z} \\
&= \mathbb{E}_Q[\log \frac{P(C, \mathbf{Z}|G)}{Q(\mathbf{Z})}] \\
&\quad (\textit{using the property of conditional probabilities}) \\
&= \mathbb{E}_Q[\log \frac{P(C|\mathbf{Z}, G) \cdot P(\mathbf{Z}|G)}{Q(\mathbf{Z})}] \\
&= \mathbb{E}_Q[\log P(C|\mathbf{Z}, G)] - \mathbb{E}_Q[\log \frac{Q(\mathbf{Z})}{P(\mathbf{Z}|G)}] \\
&\quad (\textit{using the definition of KL-divergence}) \\
&= \mathbb{E}_Q[\log P(C|G, \mathbf{Z})] - \mathrm{KL}(Q(\mathbf{Z})\|P(\mathbf{Z}|G))
\end{aligned}
\tag{26}
$$

In this manner, we can optimize the VGAE by maximizing the derived ELBO. Since we aim to perform constrained variational generation, we still need to add the constraint from the supervision of class labels. Therefore, by introducing $\lambda$ to control the importance of the constraint, the CVG objective with the VGAE architecture can be formulated as follows:

$$
\max \quad \mathbb{E}_Q[\log P(C|G, Z)] - \mathrm{KL}(Q(Z)\|P(Z|G)) - \lambda \mathrm{KL}\left(P(Y|G)\|P(Y|C)\right). \tag{27}
$$

$\square$

## C  EXPERIMENTAL SETTINGS

In this section, we introduce the detailed settings and the datasets used in our experiments. Our code is provided at https://github.com/AnonymousSubmissionPaper/CVG.

### C.1  DATASET STATISTICS AND DETAILS

In this subsection, we introduce the detailed statistics and the creation process for each dataset. Specifically, for the generation process of SP-Motif, MNIST-75sp, Graph-SST2, and Molhiv, we follow the settings in DIR (Wu et al., 2022c) to keep consistency. We create a novel dataset SP-Motif-Cor following the idea of SP-Motif.

- **SP-Motif** (Spurious-Motif) (Ying et al., 2019): This is a synthetic dataset that consists of 18,000 graphs. Each graph is composed of a base (Tree, Ladder, Wheel denoted by $S = 0, 1, 2,$, respectively) and a motif (Cycle, House, Crane denoted by $C = 0, 1, 2,$, respectively). The true label $Y$ is solely determined by the motif $C$. In the training set, we introduce false relations of varying degrees between the base $S$ and the label $Y$. Specifically, each motif is sampled from a uniform distribution, while the distribution of its base is determined by $P(S) = b \times \mathbb{I}(S = C) + (1 - b)/2 \times \mathbb{I}(S \neq C)$. We manipulate the parameter $b$ to create Spurious-Motif datasets with distinct biases. In the testing set, motifs, and bases are randomly attached to each other, and we include graphs with large bases to magnify the distribution gaps.

- **MNIST-75sp** (Knyazev et al., 2019): This dataset converts MNIST images into 70,000 superpixel graphs, with each graph containing at most 75 nodes. Superpixels represent nodes, while edges denote the spatial distance between the nodes. Each graph is labeled into one of 10 classes, and random noises are added to nodes' features in the testing set.

- **Graph-SST2** (Socher et al., 2013; Yuan et al., 2022): Graph-SST2 comprises graphs labeled according to sentence sentiment. More specifically, nodes represent tokens, and edges indicate node relations. Graphs are partitioned into different sets based on their average node degree to induce dataset shifts.

- **Molhiv** (OGBG-Molhiv) (Wu et al., 2018; Hu et al., 2020; 2021): This dataset is designed for molecular property prediction and contains molecule graphs. Specifically, nodes represent atoms, and edges represent chemical bonds. Each graph is labeled based on whether a molecule inhibits HIV replication or not.

- **SP-Motif-Cor**: We create this synthetic dataset to manually inject various degrees of intra-graph correlations for evaluation. Specifically, we keep the same number of graphs in different subsets as SP-Motif. To inject correlations for each graph, given a defined bias degree $b$, we set the features of 50% randomly selected nodes on its base graph (i.e., Tree, Ladder, or Wheel) to the same values as $Y$ (i.e., 0, 1, 2). For others, the probability is set to $(1 - b)/2$ for the other two labels. We only alter the features on 50% nodes to avoid the model naively learning classification from averaging all node features. Note that the features are only injected into the base graph. Therefore, existing methods that extract invariant subgraphs (i.e., motifs in this case) inevitably involve nodes in the base graph, which is spurious for classification based on the motif. As a result, extracting such correlated and spurious nodes will adversely impact the classification performance based on motifs.

## C.2 BASIC SETTINGS

In this subsection, we introduce the basic settings in our experiments.

**Backbone Settings.** Specifically, the classifier $f(\cdot)$ in our framework consists of a GNN and an MLP. For the GNN, we leverage a 3-layer GCN (Kipf & Welling, 2016) with a hidden size of 128, except for SP-Motif, on which we use a hidden size of 64. The dropout rate on all datasets is 0.1. The GNN is followed by an MLP to calculate the final prediction for the specific label of a graph. For the encoder GNN in our generator $g(\cdot)$, we follow the setting sin DIR (Wu et al., 2022c) and use various GNNs for different datasets. The activation functions are all set as the ReLU function.

**Training Settings.** For the training, we conduct all experiments on a NVIDIA A6000 GPU with 48GB memory. We utilize the batched GNN pipeline and set the batch size as 32 for all datasets. The learning rate is set as $10^{-3}$. For optimization, we use the Adam optimizer (Kingma & Ba, 2015) with a weight decay rate of 0. We follow the specific dataset split used in DIR. For all datasets, we train our framework for 500 epochs. We report the performance of the model obtained on the epoch that achieves the best validation performance. All the experiments are implemented with PyTorch (Paszke et al., 2019) under a BSD-style license. The required packages are listed below.

- Python == 3.7.10
- torch == 1.8.1
- torch-cluster == 1.5.9
- torch-scatter == 2.0.6
- torch-sparse == 0.6.9
- torch-geometric == 1.4.1
- numpy == 1.18.5
- scipy == 1.5.3

## C.3 HYPER-PARAMETER SETTINGS

In this section, we introduce the detailed parameter settings for our experiments. Specifically, we set the number of nodes in $C$ as $N = 5$. For the regularization loss weight $\beta_r$ and the generation loss $\beta_g$, we both set them as 1. The $\epsilon$ in the regularization loss is set as 0.7 for SP-Motif and 0.5 for other datasets. The number of positive samples $S$ and the number of negative samples $K$ in the generation loss are set as 1 and 5, respectively. The temperature is set as 1.

# D  EXPERIMENTAL ANALYSIS

**Quality of Generated Graphs.** Here we first provide statistics (average number of edges) of generated graphs with different losses removed, such that we can explore the relationship between the generation quality and performance. From results presented in Table 2, we can observe that with different components removed, the statistics of generated graphs will also change and influence the performance. Specifically, the generation loss results in a denser generated graph. As generation loss tends to enforce graph representations to be similar, it will more easily lead to denser graphs, of which the representations learned by GNNs

Table 2: The statistics of generated graphs and the corresponding performance (accuracy in %).

| # Nodes | $N = 5$ | | $N = 10$ | |
|---|---|---|---|---|
| Method | # Edges | Acc. | # Edges | Acc. |
| CVG w/o $\mathcal{L}_g$ | 3.74 | 70.08 | 16.25 | 69.30 |
| CVG w/o $\mathcal{L}_r$ | 12.81 | 68.96 | 37.28 | 67.58 |
| CVG | 5.68 | 76.57 | 22.17 | 76.27 |

will be similar. The regularization loss will lead to a sparse graph, as the regularization term makes edges more evenly distributed. In concrete, our designed losses based on the GIB principle can affect the generalization performance in various aspects.

**Relationship between Structures and Labels.** Moreover, we provide statistics (degree distribution) about the relationship between label information and the generated graphs on the SP-Motif dataset. Note that as we use a weight matrix for each node (index) in the generated graph, they maintain distinct information. From the results presented in Table 3, we can observe that for generated graphs of different labels, the degree distribution is distinct. For example, in generated graphs of label 1 (cycle), most degrees are distributed

Table 3: The degree distribution of each node on the generated graphs regarding different labels.

| Node Index | 1 | 2 | 3 | 4 | 5 |
|---|---|---|---|---|---|
| Label 1 (Cycle) | 3.21 | 0.35 | 0.22 | 0.31 | 2.05 |
| Label 2 (House) | 1.10 | 1.25 | 0.98 | 0.15 | 0.13 |
| Label 3 (Crane) | 0.20 | 0.05 | 3.55 | 3.32 | 0.18 |

on node 1 and node 5. For label 2 (house), the degrees are more averagely distributed among node 1, node 2, and node 3. That being said, our framework can indeed capture the crucial label information and encode it in the graph structures.

**Visualization.** In addition to the quantitative evaluation, we also provide visualizations as examples for interpreting the structures of generated graphs. We present a showcase from the test set in the dataset SP-Motif while highlighting the important nodes, i.e., nodes with significantly higher average degrees. From the visualization presented in Fig. 7, we observe that for generated graphs of various labels, the structures are also different, represented by the high degrees of specific nodes (e.g., node 1 and node 5 for label 1). The results align with Table 3, where we find the distinct degree distributions regarding different labels. The visualization results indicate that our strategy encodes label information within specific structures in the generated graphs.

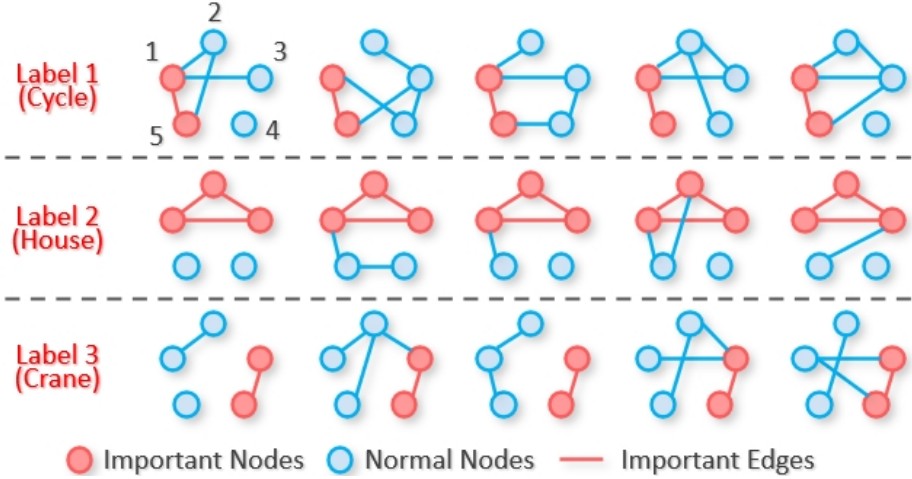

Figure 7: Visualizations of the generated graphs for samples of various labels from the testing set in the SP-Motif dataset.

