# OpenReview forum: "Constrained Variational Generation for Generalizable Graph Learning"
_ICLR.cc/2024/Conference — ICLR 2024 Conference Desk Rejected Submission_

### Official Review · Reviewer_Vcij · 2023-10-26

**Soundness:** 3 good
**Presentation:** 3 good
**Contribution:** 2 fair
**Rating:** 6
**Confidence:** 4

**Summary:**

This paper proposes a Constrained Variational Generation (CVG) framework to for graph OOD generalization.

**Strengths:**

1. The proposed Constrained Variational Generation is somewhat interesting.
2. The paper is well organized and clear.

**Weaknesses:**

1. The generated graphs may have different semantics or non-causal components. How to ensure the quality of generation? How does the quality of generation affect your performance?
2. The quality of generated graphs may be low when data is scarce.
3. The paper lacks visualization of generated graphs and training time compared to various baselines.

**Questions:**

Can you provide an evaluation of the generation quality of your model and how the quality affects the generalization performance?

---

> ### Author Response · Authors · 2023-11-17
> **Response to Reviewer Vcij (Part I)**
>
> Dear Reviewer Vcij:
>
> &nbsp;
>
>
>
> Thank you for your insightful feedback on our paper. We highly value your comments, as they have provided us with an opportunity to enhance the clarity and depth of our research. Please find below our responses to your concerns:
>
> &nbsp;
>
>
> > W1: The generated graphs may have different semantics or non-causal components. How to ensure the quality of generation? How does the quality of generation affect your performance?
>
> A1:
> (1) In our framework, the quality of the generation is **ensured by our optimization strategy** based on the Graph Information Bottleneck (GIB) principle. Particularly, we derive three distinct losses based on the GIB principle, which involves information from multiple sources, including labels, regularization, and similarity to other generated graphs. These losses help ensure the quality of generation.
>
> (2) Regarding how the quality would affect the performance, we empirically evaluate the **effectiveness of these losses** as presented in Fig. 4. Here we provide statistics (average number of edges) of generated graphs on SP-Motif with different losses removed, such that we can explore the relationship between the generation quality and performance.
>
> | Variant   | # Edges (N=5) | Acc. (N=5) | # Edges (N=10) | Acc. (N=10) |
> |-----------|---------------|------------|----------------|-------------|
> | CVG w/o $\mathcal{L}_g$ |3.74           | 70.08      | 16.25        | 69.30       |
> | CVG w/o $\mathcal{L}_r$ |   12.81  | 68.96      |    37.28    | 67.58       |
> | CVG        | 5.68         | 76.57      | 22.17        | 76.27       |
>
> *The statistics of generated graphs and the corresponding performance (accuracy in %).*
>
>
> From the results, we can observe that with different components removed, the statistics of generated graphs will also change and **influence the performance**. Specifically, the generation loss results in a **denser** generated graph. As generation loss tends to enforce graph representations to be similar, it will more easily lead to denser graphs, of which the representations learned by GNNs will be similar. The regularization loss will lead to a **sparser** graph, as the regularization term makes edges more evenly distributed. In concrete, our designed losses based on the GIB principle can affect the generalization performance in various aspects. We also include this part in the newly added **Appendix D**.
>
>
> &nbsp;
>
>
> > W2: The quality of generated graphs may be low when data is scarce.
>
> A2: As our generation process is only based on the original graph (no additional information involved), the quality is not directly affected by scarce data. Nevertheless, during training, the amount of training data can indeed affect the optimization results. Here we provide results with various training data ratios to explore the robustness of our work **with limited training data**.
>
>
> | Method     | SP-Motif | SP-Motif (50%) | SP-Motif (10%) | Molhiv |Molhiv (50%)| Molhiv (10%)|
> |------------|---------------------|------------------|------------------|------------------|------------|--------|
> | DIR        | 47.03               | 41.07            | 37.08                    | 77.05      | 75.18      | 71.02  |
> | GIL        | 55.44               | 52.54            | 47.90                  | 79.08      | 77.05    | 74.57  |
> | CIGA       | 76.52               | 70.57            | 65.71              | 79.75      | 75.39      | 71.80  |
> | GSAT      | 74.95               | 69.27            | 63.87              | 80.67     | 77.28      | 74.50  |
> | **CVG (Ours)** | **79.62**           | **74.57**        | **69.16**         |**81.09**   | **80.21**  | **76.35** |
>
> *The graph OOD generalization results (Test accuracy in % for SP-Motif and ROC-AUC for Molhiv). The best results are in **bold**.*
>
>
> From the results, we can observe that our framework still preserves generalizability even with limited training data. This is attributed to our designed GIB-based optimization strategy that can effectively learn label information from various distributions.
>
>
>
> &nbsp;

---

> ### Author Response · Authors · 2023-11-17
> **Response to Reviewer Vcij (Part II)**
>
> > W3: The paper lacks visualization of generated graphs and training time compared to various baselines.
>
>
> A3: Thank you for pointing out the necessity of including visualizations of generated graphs and training time.
>
> (1) Regarding the generated graphs, we do not originally include the visualizations because the generated graphs contain information in **both structures and node features**. Nevertheless, we find that in the generated structures, there still exists interpretable information. In **Appendix D**, We provide the **visualization** of several generated graphs as examples for interpreting the structures of generated graphs. Specifically, we present a showcase from the test set in the dataset SP-Motif while highlighting the important nodes, i.e., nodes with significantly higher average degrees. The visualization results indicate that our strategy encodes label information within specific structures in the generated graphs.
>
>
>
> (2) Regarding the training time, we apologize for not including the comparison to baselines in our paper. Here we provide experimental results in terms of training time on various methods, trained on a NVIDIA A6000 GPU, as follows:
>
> | Method     | SP-Motif (Balanced) | MNIST-75sp | Graph-SST2 | Molhiv |
> |------------|---------------------|------------------|------------------|------------------|
> | DIR        | 264.8               |            367.5    | 243.0      | 1234.5  |
> | GIL        | 358.9              |          668.7      | 811.0     | 1909.2  |
> | CIGA       | 197.2            |            417.5    | 433.7    |1028.9  |
> | GSAT      | 177.2              |         358.7    | 972.6      | 1534.8|
> | **CVG (Ours)** | **125.8**          | **167.2**  | **187.8**  | **687.4** |
>
> *Training time (second) of various methods on four datasets.*
>
> From the training time comparison, we find that our method is consistently outperforming other methods in terms of training time. Particularly, our method is more efficient on the MNIST-75sp dataset, which has a significantly larger graph size. This is because our method aims to generate graphs instead of extracting subgraphs, and thus is more computationally efficient on input graphs of large sizes.

---

> ### Author Response · Authors · 2023-11-21
> **Looking Forward to Your Feedback**
>
> Dear Reviewer Vcij,
>
> Thank you so much for your advice! We have responded to your questions and concerns.
>
> We are willing to answer any further questions. Looking forward to your feedback!
>
> Thank you!
>
> Best,
> Authors

---

> ### Author Response · Authors · 2023-11-22
> **Appreciating Your Feedback**
>
> Dear Reviewer Vcij,
>
> Thank you so much for your advice. We hope that our answers can address your concerns.
>
> We are looking forward to your feedback!
>
> Thank you!
>
> Best,
> Authors

---

> > ### Comment · Reviewer_Vcij · 2023-11-23
> > **Response to rebuttal**
> >
> > The authors have addressed most of my concerns, and I will raise my assessment. However, with regard to the generated graphs visualized in Figure 7, it seems that the results are not very meaningful (e.g. for label 2, it generates triangles instead of houses).

---

> > > ### Author Response · Authors · 2023-11-23
> > > **Thank you so much!**
> > >
> > > Dear Reviewer Vcij,
> > >
> > > Thank you so much for your recognition of your work! We really appreciate your constructive advice.
> > >
> > > We agree with you that the generated graphs do not exactly resemble the specific labels (e.g., houses). Instead, our framework aims to generate **a specific structure** for each label. For example, our framework will generate triangles for input graphs with houses, and classify them as Label 2 based on the triangles. We will provide a more detailed discussion in the revised version.
> > >
> > > Thank you so much for your effort!
> > >
> > > Best,
> > > Authors

---

### Official Review · Reviewer_Vto5 · 2023-10-31

**Soundness:** 3 good
**Presentation:** 3 good
**Contribution:** 2 fair
**Rating:** 6
**Confidence:** 4

**Summary:**

The paper proposes a novel framework for out-of-distribution (OOD) generalization on graph data. The framework uses a constrained variational generation (CVG) approach, which generates new graphs for classification based on the original graphs and a latent variable. The generation is optimized under the guidance of the graph information bottleneck (GIB) principle, which balances the trade-off between label relevance and distribution invariance. The paper leverages the variational graph auto-encoder (VGAE) structure as the generator and provides theoretical analysis to justify the GIB-based objective. The paper also conducts extensive experiments on synthetic and real-world graph datasets and demonstrates the superiority of the framework over state-of-the-art baselines.

**Strengths:**

1. This paper is well-written. The proposed technique is generally sound. Theoretical analysis and proofs are provided as necessary.
2. The idea of generating graphs under the guidance of the Graph Information Bottleneck principle is novel and interesting.
3. The experimental results are impressive, outperforming the baselines.

**Weaknesses:**

1. In the Introduction section, the authors discuss certain challenges. Are there theoretical or empirical evidences that suggest existing subgraph extraction-based methods struggle to "identify distinct invariant subgraphs" or "encounter significant limitations during substantial distribution shifts"?
2. The idea is simple, while the motivation is not well discussed. Compared with the graph information bottleneck method, the authors try to generate graphs rather than continual representations to learn essential information correlated with labels. I may be concerned about what is the benefits of this additional step, in other words, why the generated discrete graphs are better than continual representations in retaining invariant information.
3. The experimental results appear somewhat constrained, with only four graph-level datasets. More extensive experimental analysis could be helpful. For example, how are the generated graphs relevant to the label information, can the noised edges be prevented in the generated graphs? Especially for the SP-Motif dataset, more cases and illustrations would be better.

**Questions:**

1. Explain what is the invariant subgraph exactly.
2. How is the out-of-distribution (OOD) setting configured for the test data across various datasets in the experiments? It is an important setting that should be included in the main part of the manuscript.
3. The authors evaluate the proposed methods on graph classification tasks. Can the methods be applied to node-level tasks?

---

> ### Author Response · Authors · 2023-11-17
> **Response to Reviewer Vto5 (Part I)**
>
> Dear Reviewer Vto5:
>
>
> &nbsp;
>
>
>
> Thank you for your valuable feedback on our paper. Your insights have greatly helped in refining our work. Below, we address each of your concerns and the steps we have taken to improve our work:
>
>
> &nbsp;
>
>
>
> > W1: Are there theoretical or empirical evidences that suggest existing subgraph extraction-based methods struggle to "identify distinct invariant subgraphs" or "encounter significant limitations during substantial distribution shifts"?
>
> A1: Yes, we already provide **empirical evidence** for such a phenomenon using datasets SP-Motif and SP-Motif-Cor (created by us), as illustrated in Fig. 4 of our paper. From the results, we observe that existing methods are less competitive when the intra-graph correlations increase (i.e., more difficult to identify distinct invariant subgraphs) and inter-graph distinctions are larger (i.e., larger distribution shifts).
>
> Our intuition is that, in real-world datasets (e.g., Molhiv),  the shared information may reside in graph structures of several nodes and features of other nodes. In this case, solely extracting subgraphs can be suboptimal. Extraction-based methods [1-4] typically use the SP-Motif dataset, which is created by combining distinct graph structures, to showcase the effective extraction of invariant subgraphs. As SP-Motif is created and labeled based on structures and does not involve node features, we believe it is inappropriate to use it to verify the effectiveness of extraction-based methods. Therefore,
>
>
>
> &nbsp;
>
>
>
> > W2: What are the benefits of this additional step, in other words, why the generated discrete graphs are better than continual representations in retaining invariant information?
>
> A2: We propose to generate graphs to maximally preserve **intra-graph correlation information**, which instead will be lost through extracting subgraphs. We believe that discrete graphs, due to their structural nature, are more effective in **retaining such information pertinent to labels**, especially in scenarios with complex graph structures. Moreover, by generating graphs, we can leverage the **Graph Information Bottleneck (GIB) principle** to guide the optimization of our framework, with its effectiveness verified in our experiments.
>
> Note that if we remove the generated structures and only utilize continuous latent representations, such a strategy lacks **variation to preserve robustness**, and the efficacy of using variation is validated in Fig. 6. If we keep variation and do not generate structures, the case equals $N=1$. However, as shown in Fig. 5, when $N=3$, the performance already degrades significantly. Thus, our strategy of sampling multiple times from the latent representation is crucial. In concrete, by generating graphs, we can introduce more variation, which is crucial for enhancing model robustness and generalizability. Here we provide the results of only using the latent representations.
> | Method     | SP-Motif (Balanced) | MNIST-75sp | Graph-SST2 | Molhiv |
> |------------|---------------------|------------------|------------------|------------------|
> | CVG latent ($N=1$)       | 70.17            |           21.63    | 77.28      | 76.23  |
> | CVG $N=3$         | 75.68              |          27.58      | 81.60    | 79.04|
> | **CVG $N=5$** | **79.62**          | **30.12**  | **84.21**  | **81.09** |
>
> *The graph OOD generalization results (Test accuracy in \% for SP-Motif, MNIST-75sp, and Graph-SST2, ROC-AUC for Molhiv). The best results are in **bold**.*
>
> From the results, we observe that when only using the latent representations, the performance degrades greatly. The performance also drops when N (the number of nods) decreases, indicating the significance of using generated graph structures that involve multiple nodes.
>
>
> &nbsp;

---

> ### Author Response · Authors · 2023-11-17
> **Response to Reviewer Vto5 (Part II)**
>
> > W3: More extensive experimental analysis could be helpful. For example, how are the generated graphs relevant to the label information, can the noised edges be prevented in the generated graphs?
>
> A3: Thank you for your suggestion on more experimental analysis. Here we provide statistics (degree distribution) about the relationship between label information and the generated graphs on the SP-Motif dataset. Note that as we use a weight matrix for each node (index) in the generated graph, they maintain distinct information for individual node.
>
>
>
> | Node Index |    1 |    2 |    3 |    4 |    5 |
> |------------|------|------|------|------|------|
> | Label 1  (Cycle)  | 3.21 | 0.35 | 0.22 | 0.31 | 2.05 |
> | Label 2  (House)  | 1.10 | 1.25 | 0.98 | 0.15 | 0.13 |
> | Label 3 (Crane)   | 0.20 | 0.05 | 3.55 | 3.32 | 0.18 |
>
> *The degree distribution of each node on the generated graphs regarding different labels.*
>
> From the results, we can observe that for generated graphs of different labels, the degree distribution is distinct. For example, in generated graphs of label 1 (cycle), most degrees are distributed on node 1 and node 5. For label 2 (house), the degrees are more averagely distributed among node 1, node 2, and node 3. That being said, our framework can indeed capture the crucial label information and encode it in the graph structures. We also include this part in the newly added Appendix D.
>
>
>
> &nbsp;
>
>
>
>
> >Q1: Explain what is the invariant subgraph exactly.
>
> A1: The invariant subgraph is a part of the original graph and shares **invariant information** (e.g., decisive information for labels) across various distributions. For example, in SP-Motif, the labels are solely dependent on the motifs on each graph, and such motifs are the optimal invariant subgraphs that should be extracted for classification. However, real-world datasets are much different from SP-Motif. Generally, such optimal invariant subgraphs do not exist, and thus we propose to **generate graphs** instead of extracting subgraphs.
>
>
>
> &nbsp;
>
>
>
> > Q2: How is the out-of-distribution (OOD) setting configured for the test data across various datasets in the experiments?
>
> A2: The details of the OOD settings are described in Appendix C.1. We agree that these settings are crucial and apologize for not including them in the main manuscript. We have revised this part in the new version.
>
>
> &nbsp;
>
>
>
> > Q3: Can the methods be applied to node-level tasks?
>
> A3: Yes, our method is **applicable to node-level tasks**. We conduct experiments on node-level OOD generalization tasks using datasets and settings proposed by EERM [5], and provide the results as follows. To accommodate node-level tasks, we use the local subgraphs of each node as input for generation. We additionally compare our framework with more recent baselines LiSA [6], IS-GIB [7], and MARIO [8]. Results of LiSA are obtained from the paper, while IS-GIB  and MARIO are run by us.
>
>
> | Dataset | FB-100 | Twitch | Elliptic | Arxiv |
> |---------|--------|--------|----------|-------|
> | ERM | 52.75±0.63 | 52.22±0.87 | 61.60±1.11 | 44.84±1.39 |
> | EERM | 54.32±1.42 | 54.06±0.88 | 61.86±1.06 | 47.76±1.39 |
> | LiSA | 54.24±1.03 | 55.83±2.21 | 67.87±2.57 | 44.74±0.38 |
> | IS-GIB | 54.56±1.20 | 55.97±1.15 | 70.53±1.35 | 48.80±1.64 |
> | MARIO | 53.87±1.37 | 55.10±1.86 | 71.22±2.01 | 49.35±2.76 |
> | **CVG** | **55.04±1.67** | **57.03±1.72** | **81.09±1.36** | **50.58±1.83** |
>
> *The node-level OOD generalization results (test accuracy in % for FB-100 and Arxiv, F1 score in % for Elliptic, and ROC-AUC in % for Twitch). The best results are in **bold**.*
>
> From the results, we can observe that our framework CVG still outperforms other baselines, indicating the applicability of CVG on various tasks. In the future, we will also explore the potential of our framework for node-level tasks.
>
> &nbsp;
>
>
>
> [1] Miao, Siqi, Mia Liu, and Pan Li. "Interpretable and generalizable graph learning via stochastic attention mechanism." ICML 2022.
> [2] Wu, Ying-Xin, et al. "Discovering invariant rationales for graph neural networks." ICLR 2022.
> [3] Li, Haoyang, et al. "Learning invariant graph representations for out-of-distribution generalization." NeurIPS 2022.
> [4] Chen, Yongqiang, et al. "Learning causally invariant representations for out-of-distribution generalization on graphs." NeurIPS 2022.
> [5] Wu, Qitian, et al. "Handling distribution shifts on graphs: An invariance perspective."  ICLR 2022.
> [6] Yu, Junchi, Jian Liang, and Ran He. "Mind the Label Shift of Augmentation-based Graph OOD Generalization." CVPR 2023.
> [7] Yang, Ling, et al. "Individual and structural graph information bottlenecks for out-of-distribution generalization." IEEE TKDE 2023.
> [8] Zhu, Yun, et al. "MARIO: Model Agnostic Recipe for Improving OOD Generalization of Graph Contrastive Learning." arXiv 2023.

---

> > ### Comment · Reviewer_Vto5 · 2023-11-21
> >
> > The author has addressed most of my concerns, so I will change my rating to positive.

---

> ### Author Response · Authors · 2023-11-21
> **Thank you!**
>
> Dear Reviewer Vto5,
>
> Thank you so much! We really appreciate your suggestions and would like to include the analysis in the paper. We will keep up the good work!
>
> Thank you!
>
> Best,
> Authors

---

### Official Review · Reviewer_nauu · 2023-11-04

**Soundness:** 3 good
**Presentation:** 3 good
**Contribution:** 3 good
**Rating:** 8
**Confidence:** 3

**Summary:**

The paper addresses the problem of Out of distribution generalization  on graph structured data. General strategies for OOD problem is extracting the invariant subgraphs which most informative of the class labels. However, the authors identify, such techniques fail to adequately address the challenges of intra-graph correlations, wherein partial correlations are present which may not be invariant and inter-graph distinctions, wherein invariant substructures may not be informative for the class labels.  The authors propose to address these challenges by devising a generative framework instead of subgraph extraction, while preserving the class label information. To achieve this, they optimize a approximate variational objective with the principle of the graph information bottleneck. The experiments indicate the effectiveness of the approach.

**Strengths:**

1. The paper is well-motivated and the presented idea is very interesting.
1. The development of the method is principled, optimizing the conditional variational objective with suitable approximations.
1. The paper is well-written and easy to follow. The presentation of the idea flows coherently.
1. The experiments are interesting. However, it would be further improved with additional visualizations of the generated graphs.

**Weaknesses:**

1. It would be helpful to visualize a set of generated graphs vs the actual graphs to find if there exists any structural similarity or if the model is learning any structural information relevant to the task. With SP-motif dataset, we should be able to see the generated graphs and contrast it with the actual motifs relevant to the class label.
1. Are there any relations to the graph structures generated in C with the class labels? Do we see similar C graphs for same class labels?
1. The learning can be made stronger if we have supervision for generation. One of the datasets, SP-Motif, does contain the actual supervision for C. It would be interesting to see if there gains to be made with supervision signals instead of the self-supervised loss in generation.


In summary, the proposed method is interesting and seems to make reasonable contributions to the study of OOD generalizations. The paper can be strengthened wiIth further analysis on the generated outputs.

**Questions:**

Please address the weaknesses.

---

> ### Author Response · Authors · 2023-11-17
> **Response to Reviewer nauu**
>
> Dear Reviewer nauu:
>
> &nbsp;
>
>
> Thanks for your insightful feedback and recognition of our work! Here we would like to address your questions as follows:
>
>
> &nbsp;
>
>
>
> > W1: It would be helpful to visualize a set of generated graphs vs the actual graphs to find if there exists any structural similarity
>
> A1: Thank you for your precious suggestion. We added several **case studies** to illustrate the quality of the generation in the newly added Appendix D. In particular, our framework does not require the generated graphs to be similar to the actual graphs. Instead, we hope to maximally encode label information in the generated graphs while filtering out irrelevant information based on the GIB principle. The added case studies demonstrate how our framework contributes to the generalizability.
>
> &nbsp;
>
>
> > W2: Are there any relations to the graph structures generated in C with the class labels? Do we see similar C graphs for same class labels?
>
> A2: During training, with our supervision loss based on the GIB principle, our framework is optimized to generate graphs that maximally **preserve label information**. Therefore, the generated graphs with the same labels are also more similar.
>
>
> &nbsp;
>
>
> > W3: The learning can be made stronger if we have supervision for generation.
>
> A3: Thank you for your suggestion! We agree that with actual supervision of specific graph structures, the generation can be more precise and helpful. A feasible implementation would be enforcing the generated graphs to be as similar as possible to the actual supervision. We will further explore the potential of this strategy in future work.

---

### Official Review · Reviewer_ZRDy · 2023-11-06

**Soundness:** 3 good
**Presentation:** 3 good
**Contribution:** 3 good
**Rating:** 6
**Confidence:** 4

**Summary:**

This work aims at resolving the OOD graph classification problem from the perspective of graph generation from the observed graphs. The proposed method consists of three modules, including graph generation from latent variable Z following Gaussian distribution, and two regularization terms over the generated graphs by grounding node representations as Gaussian and edge as Bernoulli distributions. By deriving the variational lower bounds of of graph information bottleneck, the proposed method tends to generate diverse graphs constraint to the observed ones. Overall, it's a good paper to read and experimental show the superiority of the work.

**Strengths:**

Strengths:
1. This work points out the limitations of invariant learning methods for OOD graph classification task, and proposes to deal with this problem by generating graphs that potentially are keeping similar label distributions but with different structure.
2. Employing graph information bottleneck helps to derive the optimization objective and the constraints over the generated graphs.
3. Experimental results demonstrate the superiority of the proposed method.

**Weaknesses:**

Weaknesses:
1. The motivation behind this work deserves more words. It's not clear why we can tackle the limitations of invariant learning with generated graphs. It's just a descriptive property that is not sufficient enough to tell which part of the component contributes the experimental improvement.
2. It's lack of case studies to show what kinds of generated graphs that exist in test but not in training data. It seems like the generation task just serves as a regularization but not for generating meaningful graphs or molecular structure.
3. The presentation still needs a further improvement. Some notations are not clear enough. For example, in Equation (13), should $\beta_{ij}$ be $\alpha_{ij}$? If not, please give a clear justification. In Equation (14), it's not clear about how to get the hidden representations of C and C_i. As we can see that f() maps the graph representations to class distribution, then what do you mean the representations of C and C_i are learned by f()?

**Questions:**

please refer to the above mentioned points.

---

> ### Author Response · Authors · 2023-11-17
> **Response to Reviewer ZRDy**
>
> Dear Reviewer ZRDy:
>
> &nbsp;
>
> Thank you so much for your insightful suggestions! Here we would like to provide responses to your comments as follows:
>
>
>
> &nbsp;
>
>
> > W1: The motivation behind this work deserves more words. It's not clear why we can tackle the limitations of invariant learning with generated graphs.
>
> A1: The motivation of our work originates from the disadvantages of the prevalent extraction-based approaches. Instead of extracting invariant subgraphs, we aim to **preserve information on the entire graph**, and thus propose to generate new graphs from the original graph. Through generation, we can leverage the **Graph Information Bottleneck (GIB) principle** to guide the optimization of our framework, thereby further improving the generalization performance.
>
> &nbsp;
>
> > W2: It's lack of case studies to show what kinds of generated graphs that exist in test but not in training data.It seems like the generation task just serves as a regularization but not for generating meaningful graphs or molecular structure.
>
> A2: Thank you for your precious suggestion. In the newly added Appendix D, we added several **case studies** showcasing various instances where our model successfully generates meaningful graphs that are not present in the training set but appear in the test set.
>
> We formulate our framework as a generation task for two reasons: (1) We aim to introduce more **variation**, and the benefits are empirically verified in Fig. 6. (2) Generating graphs enables us to leverage the **GIB principle** for optimization, thereby enhancing the generalizability. In concrete, although our generated graphs are not necessarily the same as the invariant structures, the generation can maximally preserve classification information from the original graph while filtering out irrelevant information based on the GIB principle.
>
> &nbsp;
>
> > W3: Some notations are not clear enough in Eq. (13) and Eq. (14).
> A3: Thank you so much for pointing out the errors in the notations.
> 1. Regarding Eq. (13), the $\beta_{i,j}$ in Eq. (13) should be $\alpha_{i,j}$, which denotes the edge weight between node $i$ and $j$.
> 2. Regarding Eq. (14), as stated in Appendix C.2, our classifier $f(\cdot)$ consists of a GNN for learning graph representations and an MLP to make predictions after the GNN. Here the hidden representations refer to the outputs by the GNN, which are 128-dimensional vectors. We apologize for the confusion here, and we will revise it in the new version.

---

### Author Response · Authors · 2023-11-23

Dear Reviewers,

Thank you so much for your efforts and constructive comments. The discussion will end soon. We are more than happy to address your further concerns and revise our paper accordingly.

Best,
Authors